# Zika Virus-Infected Monocyte Exosomes Mediate Cell-to-Cell Viral Transmission

**DOI:** 10.3390/cells13020144

**Published:** 2024-01-12

**Authors:** Pedro Pablo Martínez-Rojas, Verónica Monroy-Martínez, Lourdes Teresa Agredano-Moreno, Luis Felipe Jiménez-García, Blanca H. Ruiz-Ordaz

**Affiliations:** 1Departamento de Biología Molecular y Biotecnología, Instituto de Investigaciones Biomédicas, Universidad Nacional Autónoma de México, Ciudad de México 04510, México; pedropablo.martinezrojas@gmail.com (P.P.M.-R.); vmonroy@iibiomedicas.unam.mx (V.M.-M.); 2Departamento de Biología Celular, Facultad de Ciencias, Universidad Nacional Autónoma de México, Ciudad de México 04510, México; agredano-moreno@ciencias.unam.mx (L.T.A.-M.); luisfelipe_jimenez@ciencias.unam.mx (L.F.J.-G.)

**Keywords:** Zika virus, extracellular vesicles, small extracellular vesicles, exosomes, human monocytes, cell-to-cell viral transmission

## Abstract

Zika fever is a reemerging arthropod-borne viral disease; however, Zika virus (ZIKV) can be transmitted by other, non-vector means. Severe Zika fever is characterized by neurological disorders, autoimmunity, or congenital Zika syndrome. Monocytes are primary ZIKV targets in humans and, in response to infection, release extracellular vesicles like exosomes. Exosomes mediate intercellular communication and are involved in the virus’s ability to circumvent the immune response, promoting pathological processes. This study aimed to evaluate the role of monocyte exosomes in cell-to-cell viral transmission. We isolated exosomes from ZIKV-infected monocytes (Mø exo ZIKV) by differential ultracentrifugation and identified them by nanoparticle tracking analysis; transmission electron microscopy; and CD63, CD81, TSG101, and Alix detection by cytofluorometry. Purified exosome isolates were obtained by uncoupling from paramagnetic beads or by treatment with UV radiation and RNase A. We found that Mø exo ZIKV carry viral RNA and E/NS1 proteins and that their interaction with naïve cells favors viral transmission, infection, and cell differentiation/activation. These data suggest that Mø exo ZIKV are an efficient alternative pathway for ZIKV infection. Knowledge of these mechanisms contributes to understanding the pathogenesis of severe disease and to the development of new vaccines and therapies.

## 1. Introduction

Zika virus (ZIKV), a single-stranded positive-sense RNA flavivirus, has reemerged as a human pathogen [1,2]. A Zika fever (ZIKF) epidemic broke out in the Americas in 2015–2016, triggering the World Health Organization (WHO) to declare a Public Health Emergency of International Concern [3]. Since then, more than 870,000 autochthonous cases have been reported on the continent, and 89 countries have reported ZIKV transmission across the Americas, Africa, Southeast Asia, and the Western Pacific [4,5,6]. ZIKF, an acute febrile disease, is caused by infection with ZIKV, which is transmitted mainly by *Aedes* spp. mosquitoes; however, non-vectorial transmission by various routes (sexual contact; from mother to child either in utero, resulting in congenital Zika syndrome (CZS), or intrapartum; transfusion of infected blood components; transplantation of infected organs; or laboratory exposure) that favor person-to-person viral spread have been described [7,8,9]. About 80% of ZIKV infections are asymptomatic, and 20% of ZIKF cases are mild. Some ZIKF cases progress to severe disease characterized by neurological disorders, autoimmunity, or thrombocytopenia. To date, no antivirals or safe vaccines are available for ZIKV infection [1,2,3,10].

Monocytes (Mø) are the primary target cells of ZIKV infection [11,12,13]. Mø constitute nearly 10% of bloodstream leukocytes, with three subpopulations: classic, intermediate, and non-classical. In infectious processes, Mø are recruited to the tissues, inducing a proinflammatory state [14,15]. Mø can be used by ZIKV as Trojan horses to infiltrate, through the vascular endothelium, to immunologically privileged tissues (brain, placenta, or testicles), contributing to severe ZIKF pathogenesis. During Mø differentiation, extracellular vesicles (EV) are released. EV are heterogeneous particles formed by the lipid bilayer, and they do not replicate [16]. According to their biogenesis, EV are classified as exosomes, which have an endosomal origin and are released by exocytosis, or as ectosomes, which are formed by budding from the plasma membrane [17]. EV carry active biomolecules, and their interactions with cells modify those cells’ behavior or phenotype. EV functions include intercellular communication, nutrient exchange, and the elimination of intracellular components [18,19]. Therefore, EV are relevant to homeostasis and disease [20].

Exosomes or small EV (sEV) are nanoparticles <200 nm in size. They can be identified by detection of tetraspanins (CD63, CD81, and CD9) or of proteins from the endosomal sorting complexes required for transport (ESCRT, like TSG101 or Alix), which are present on their membranes or as part of their cargo [16]. During infection by some RNA viruses, the viral cycle and exosome biogenesis may converge, such that viral components (genomes, antigens, or complete virions) are incorporated into the sEV content [21]. Therefore, exosomes have the potential to serve as vehicles for ZIKV transmission. Such processes have been described for other flaviviruses such as dengue virus (DENV) and West Nile virus (WNV) [22,23,24,25].

In the context of ZIKV infection, our group previously reported that exosomes from ZIKV-infected mosquito cells carry viral RNA and E protein, favoring infection and the differentiation of human Mø and endothelial cells. These cells induce the expression of damage markers and an increase in barrier permeability, and these exosomes could thus directly contribute to ZIKF pathogenesis [26]. Other studies focused on ZIKF neuropathogenesis have demonstrated that sEV from murine cortical neurons and human astrocytes/cerebral microvasculature carry viral genomic RNA and E/NS1 proteins that favor infection of naïve cells [27,28,29], showing that exosomes mediate ZIKV transmission. Studies using CZS models have demonstrated that ZIKV-infected differentiated monkey trophoblasts release exosomes containing the viral E, NS1, NS4B, and NS5 proteins, which may increase the susceptibility of other uninfected placental cells. Likewise, exosomes from ZIKV-infected neuroepithelial-like stem cells (lt-NES) are enriched with dysregulated cellular miR-4792, amiRNA associated with neurodevelopment and oxidative stress, and their transfer could affect uninfected cells, thus impairing brain development [30,31].

On the other hand, the function of exosomes released from human mononuclear cells in flavivirus pathogenesis is poorly understood. It has been reported that DENV-3-infected Mø-derived dendritic cells release exosomes that carry viral E protein and function as Trojan horses that evade the antibody response, favoring the infection of mosquito cells [25]. Also, it was found that exosomes from DENV-2-infected macrophages carry the NS3 protein, and their interaction with endothelial cells modified the physiology and integrity of these cells, inducing the release of high levels of proinflammatory cytokines [32]. The function of Mø exosomes in ZIKV infection is still unclear.

This study aimed to evaluate the potential role of exosomes released by ZIKV-infected Mø (Mø exo ZIKV) in cell-to-cell viral transmission. Mø exo ZIKV were isolated by differential ultracentrifugation and identified by nanoparticle tracking analysis, transmission electron microscopy, and cytofluorometry assays. Exosome isolates were purified from paramagnetic beads, and any viral particles or free RNA present were inactivated with UV radiation and degraded by RNase A, respectively. Our results support the conclusion that during in vitro ZIKV infection, activated intermediate Mø release exosomes that carry viral components as part of their cargo. The naïve-cell–exosomes interaction promotes viral transmission, infection, and cell differentiation/activation. Hence, the Mø exo ZIKV are an efficient alternative transmission route that may contribute to disease progression.

## 2. Materials and Methods

### 2.1. Cell Cultures and Zika Virus Strain

*Aedes albopictus* mosquito C6/36 cells (ATCC CRL-1660, USA), Vero cells (ATCC CCL-81), and monocytes THP-1 (ATCC TIB-202) were maintained in Leibovitz L15 (Biowest, Riverside, MO, USA), DMEM (Biowest), and RPMI-1640 (Biowest) media, respectively, supplemented with 10% (*v*/*v*) of fetal bovine serum (FBS; Biowest), 2 mM L-glutamine (Biowest), and antibiotics (penicillin 100 U/mL, streptomycin 0.1 mg/mL, and amphotericin B 0.25 µg/mL; Biological Industries, Cromwell, CT, USA). The Leibovitz L15 medium also contained 10% tryptose phosphate broth (DIFCO, Lawrence, KS, USA). C6/36 cells were incubated at 28 °C without CO_2_, while Vero and THP-1 cells were incubated at 37 °C in 5% CO_2_. The ZIKV MR766 strain (GenBank Accession HQ234498.1) was used.

### 2.2. Viral Propagation

Viral propagation was performed based on previous methods [33,34]. ZIKV was inoculated in C6/36 cells at a multiplicity of infection (MOI) of 0.5 for 7 days. Culture fluids were harvested by detachment through freeze-thaw thermal shock at 2 °C and by mechanical force with a cell scraper (Corning, Corning, NY, USA), then collected in sterile conical tubes (Corning). Samples were vortexed and clarified by centrifugation (GH3.8 rotor, Beckman GPR Centrifuge; Beckman Coulter, Inc., Brea, CA, USA) at 2000× *g* for 15 minutes (min) at 4 °C. Supernatants were filtered through a 0.22 µm pore (Millipore, Burlington, MA, USA), centrifuged (SW28 rotor, Beckman XL-90 Ultracentrifuge, Beckman Coulter, Inc.) at 120,000× *g* for 2 hours (h) at 4 °C, and discarded. Pellets were resuspended in 5 mL of phosphate-buffered saline (PBS) 1× and stored in 0.5 mL aliquots at −72 °C. Purity was confirmed by nanoparticle tracking analysis (NTA), detection of total viral RNA by agarose gel electrophoresis, viral RNA amplification by RT-PCR, and infectivity in Vero cells by immunofluorescence assays (Appendix A).

### 2.3. Viral Titration by Lytic Plaque Assays

Confluent Vero cell monolayers in 24-well culture plates (Corning) were each inoculated with 450 µL of viral dilution (10-fold serial dilutions) in FBS-free medium and incubated at 37 °C with 5% CO_2_ for 2 h. Next, the dilutions were removed and the monolayers were overlaid with DMEM containing 1% carboxymethylcellulose (Sigma-Aldrich, St. Louis, MO, USA) and 2.5% FBS. The cells were incubated at 37 °C with 5% CO_2_ until plaques were observed through light-field microscopy (Olympus IX71 inverted microscope; Olympus Corp., Miami, FL, USA). The cells were fixed with 96% methanol (J.T.Baker, Mexico City, Mexico) on day 7 post-infection and stained with 1% crystal violet (Sigma-Aldrich). Viral titer was expressed as plaque-forming units (PFU) per mL (Appendix A).

### 2.4. Preparation of Fetal Bovine Serum Depleted of Extracellular Vesicles (FBS-dEV)

Sterile conical tubes containing the FBS were centrifuged at 900× *g* for 10 min at 4 °C and filtered through a 0.22 µm pore. Samples were centrifuged at 120,000× *g* for 18 h at 4 °C, transferred to new sterile conical tubes, and stored at 2 °C until use [16].

### 2.5. Monocyte Viral-Infection Assay

Monocytes (Mø; 1.0 × 10^6^) were seeded in 6-well culture plates (Corning), inoculated with ZIKV (MOI 1, 3, and 5) in FBS-free medium, and incubated at 37 °C with 5% CO_2_ for 2 h. The cells were collected in sterile tubes, washed with PBS 1×, and centrifuged twice at 900× *g* at room temperature (rt) for 10 min. The cells were resuspended in RPMI-1640 medium supplemented with 5% FBS-dEV and incubated for 24, 48, 72, and 96 h.

### 2.6. Detection of Viral E Protein by Immunofluorescence (IF) Assays

Vero cells or Mø (2.5 × 10^5^) were seeded in an 8-well chamber slide (Lab-Tek II; Thermo Fisher Scientific, Waltham, MA, USA), fixed with 2% paraformaldehyde (PFA; Sigma-Aldrich) for 5 min at 4 °C, and washed with 2% bovine serum albumin (BSA; Biowest) in PBS 1×. Non-specific binding sites were blocked with 2% BSA in PBS for 30 min at rt, and the cells were then washed with 0.5% BSA in PBS. After the mouse anti-ZIKV ENV IgG1 antibody (identified as anti-E ZIKV) clone 1413267 (Appendix A) was added (1:300), the cells were incubated overnight at 4 °C, then washed with 0.5% BSA in PBS. Next, the Alexa Fluor 555-conjugated donkey anti-mouse IgG secondary antibody (Appendix A) was added (1:500). The cells were then incubated for 2 h at rt protected from light and washed with 0.5% BSA in PBS. Finally, the slide was covered with DAPI mounting medium (AAT Bioquest, Sunnyvale, CA, USA). The mouse IgG1 antibody clone P3.6.2.8.1 (Appendix A) was used as an isotype control. Observations were performed in an Olympus IX71 inverted microscope with a built-in Olympus DP72 digital camera; images were analyzed with ImageJ software version 1.50i (NIH, Bethesda, MA, USA).

### 2.7. RNA Extraction, Purification, and Quantification

The QIAamp RNA Mini kit (Qiagen, Hilden, Germany) was used as follows: samples were lysed under highly denaturing conditions with lysis buffer and 70% ethanol (J.T.Baker), vortexed at high speed, and incubated for 15 min at rt. The entire volume was placed in separation tubes containing mini-columns with RNA-adsorption membranes. Tubes were centrifuged at 2947× *g* (Eppendorf 5415 C centrifuge; Merck, Darmstadt, Germany) for 2 min at rt, and the filtrates were discarded. Then, membrane contaminants were washed with absorber waste buffers and centrifuged at 2947× *g* for 2 min at rt, and the filtrates were discarded. Finally, the elution buffer (0.4% sodium azide in RNase-free water) was added, and the tubes were centrifuged at 16,047× *g* for 2 min at rt. The RNA filtrates were collected in microcentrifuge tubes (Labcon, Petaluma, CA, USA) at 4 °C. RNA was quantified using the NanoDrop ND1000 spectrophotometer (Thermo-Fisher Scientific). Samples were used immediately or stored at −72 °C until use.

### 2.8. RNA Detection by End-Point Reverse Transcriptase-Polymerase Chain Reaction (RT-PCR)

Viral RNA amplification was performed based on previous protocols [26,35,36] using the OneStep RT-PCR kit (Qiagen). The primers for the conserved region of the E protein, ZIKV FW [5′-GCTGGDGCRGACACHGGRACT-3′] (Mfg. ID 275853243, Integrated DNA Technologies [IDT], San Diego, CA, USA) and ZIKV RV [5′-RTCYACYGCCATYTGGRCTG-3′] (Mfg. ID 275853246, IDT) were used. The 394 bp amplicon was visualized on a 2% ethidium bromide-stained 1.2% agarose gel (Thermo Fisher Scientific) using the Typhoon FLA 9500 scanner (GE Healthcare, Chicago, IL, USA).

### 2.9. Detection of Viral Proteins and Cell Markers by FACS

Monocytes (1.0 × 10^6^) were collected, centrifuged at 900× *g* for 5 min at 4 °C, and fixed with 2% PFA for 5 min at 4 °C. Non-specific binding sites were blocked with 2% BSA in PBS 1× for 30 min at rt. For conjugated antibodies (Appendix A), samples were suspended in 0.5% BSA in PBS containing the antibody (1:20), incubated for 30 min at rt while protected from light, washed with 0.5% BSA in PBS, centrifuged, and kept in 0.5% BSA in PBS at 4 °C. For primary antibodies (Appendix A), cells were suspended in 0.5% BSA in PBS containing the antibody (1:300), incubated overnight at 4 °C, washed with 0.5% BSA in PBS, and centrifuged. Secondary antibody (1:500) was added, and the cells were incubated for 2 h, washed with 0.5% BSA in PBS, centrifuged, and kept in 0.5% BSA in PBS, protected from light. The FACSCalibur flow cytometer (BD Biosciences, San Jose, CA, USA) was used for analysis.

### 2.10. Isolation of Monocyte Exosomes (Mø Exo) from Cell Culture Media by Ultracentrifugation

Cell culture media were collected in sterile conical tubes and centrifuged at 900× *g* at 4 °C for 10 min. The supernatants were transferred to other tubes, and the cell pellets were discarded. Samples were centrifuged at 2000× *g* at 4 °C for 10 min. The supernatants were transferred to sterile ultracentrifuge tubes (25 × 89 mm, Beckman Coulter, Inc.) and centrifuged at 10,000× *g* at 4 °C for 35 min, and the cell debris was discarded. The supernatants were transferred to other sterile ultracentrifuge tubes, centrifuged at 120,000× *g* at 4 °C for 70 min, and discarded. The exosome pellets were resuspended in 10 mL of PBS 1× at 4 °C, incubated at rt for 30 min with 100 rpm agitation, and filtered using a 0.22 µm pore size. After the addition of 12.5 mL of PBS, the exosome suspension was centrifuged at 120,000× *g* for 70 min at 4 °C. Supernatants without EV (Non-EV SNT) were separated into aliquots and stored at −72 °C. Exosome pellets were suspended in 5 mL of PBS at 4 °C and either collected in sterile microcentrifuge tubes to be used immediately or stored at −72 °C.

### 2.11. Exosome Quantification by Nanoparticle Tracking Analysis (NTA)

Detection of nanoparticles <200 nm was performed using NanoSight NS300 equipment (Malvern Panalytical Products, Mexico City, Mexico) under the following conditions: camera level of 14, detection limit of 2, temperature of 20 °C, acquisition time of 90 seconds (s), and 3 consecutive repetitions in different fields. Samples were diluted 1:50 in PBS 1×. Polystyrene microspheres of 100 and 200 nm (NTA4088 and NTA4089, Malvern) were used (1:500) as controls.

### 2.12. Exosome Coupling to Paramagnetic Beads for Detection of Exosomal Markers by FACS

Exosomes were coupled to anti-CD63-antibody-coated paramagnetic (pmag) beads (Catalog # 10606D, Invitrogen, Thermo Fisher Scientific). Briefly, suspensions of 100 µL of exosome isolates and 20 µL of pmag beads were added to sterile round-bottom microcentrifuge tubes (Labcon) and incubated for 18 h at 4 °C with 1000 rpm agitation. Pmag-bead-coupled exosomes were separated from the liquid matrix using the DynaMag-2 magnet (Thermo Fisher Scientific) and resuspended in 0.5% BSA in PBS 1× (Appendix A) containing the specific antibodies (Appendix A). For detection of conjugated anti-CD63 and anti-CD81 (1:20), the samples were incubated for 2 h at rt and kept in 0.5% BSA in PBS (Appendix A). For detection of TSG101 and Alix, the samples were first fixed and permeabilized with 96% methanol (J.T.Baker) for 1 min at 4 °C and blocked with 3% BSA in PBS for 1 h at rt. For detection of primary antibodies (1:300), the samples were incubated overnight at 4 °C, suspended in 0.5% BSA in PBS containing the secondary antibody (1:500), incubated for 2 h at rt, and kept in 0.5% BSA in PBS (Appendix A). The FACSCalibur flow cytometer was used for analysis.

### 2.13. Morphological Characterization of Exosomes by Transmission Electron Microscopy (TEM)

Sample preparation was performed as we described previously [26]. Briefly, exosome isolates were fixed in 2.5% glutaraldehyde (EMS, Hatfield, PA, USA) and 4% PFA (1:1) for 2 h at rt. Post-fixation was performed with 2% osmium tetroxide (Alfa Aesar, Thermo Fisher Scientific) for 90 min at rt, followed by triple washes with PBS, dehydration with ascending percentages of ethanol (30, 50, 70, 80, 90, and 96%), and triple washes in absolute ethanol (J.T.Baker) and propylene oxide (Sigma-Aldrich) at rt. Next, samples were incubated for 18 h at rt in propylene oxide and epoxy resin (1:1) and embedded in pure epoxy resin (EMS) at 60 °C for 48 h. Samples were cut into ultrathin sections (40–50 nm thickness), mounted on formvar (EMS)-covered copper grids, and contrasted with uranyl acetate (Merck) for 30 min and lead citrate (EMS) for 10 min at rt. Observations were performed in a transmission electron microscope (JEM1010, JEOL, Peabody, MA, USA) with an adapted CCD300-RC camera (DAGE-MTI, Michigan City, IN, USA).

### 2.14. ZIKV Inactivation in Exosome Isolates

Exosome isolates were treated with three cycles of UV light at 1200 µJ (×100) and RNase A (Thermo Fisher Scientific), as we described previously [26] (Appendix A). After total RNA quantification, 10 µg/mL of RNase A was added to each sample and the samples were incubated for 1 h at 37 °C. The RNase inhibitor RNaseOUT (40 U/µL; Invitrogen) was added in a 1:1 ratio with RNase A volume. Samples were incubated for 15 min at 37 °C, kept at 4 °C, and identified as Mø exo ZIKV (RNase A + UV).

### 2.15. Exosome Purification by Uncoupling from Paramagnetic Beads

We developed the following protocol for exosome recovery (Appendix A) based on the manufacturer’s indications that the binding affinity of the pmag beads for their coupled exosomes decreases with vigorous vortexing. Briefly, pmag-bead-coupled exosome suspensions were suspended in PBS 1×, homogenized by pipetting, high-speed vortexed for 2 min at rt, centrifuged at 16,047× *g* for 20 s at rt, and vortexed at high speed for 2 min at rt. After the pmag beads were removed using a magnet, the supernatants (containing uncoupled exosomes) were transferred to sterile microcentrifuge tubes, homogenized by pipetting, kept at 4 °C, and identified as purified Mø exo ZIKV (Mø exo ZIKVp).

### 2.16. Detection of Viral E and NS1 Proteins in Paramagnetic-Bead-Coupled Exosomes by FACS

Paramagnetic-bead-coupled exosomes were fixed with 2% PFA for 5 min at 4 °C, and the non-specific binding sites were blocked with 3% BSA in PBS 1× for 1 h at rt. Samples were suspended in 0.5% BSA in PBS containing the specific primary antibody (1:300) and incubated overnight at 4 °C. Samples were suspended in 0.5% BSA in PBS containing the secondary antibody (1:500), incubated for 2 h at rt while protected from light, and kept in 0.5% BSA in PBS. The FACSCalibur flow cytometer was used for analysis.

### 2.17. Detection of Viral E and NS1 Proteins by Enzyme-Linked Immunosorbent Assay (ELISA)

Exosome isolates suspended in a carbonate buffer solution (pH 9.6) were added (100 µL/well) to a flat-bottomed 96-well polystyrene plate (Thermo Fisher Scientific) and incubated overnight at 4 °C. The wells were rinsed 3 times with 0.02% Tween 20 (Sigma Aldrich) washing solution in PBS 1×. The non-specific binding sites were blocked with 3% BSA in PBS and incubated for 1 h at rt. The primary antibody (1:300) was added and incubated overnight at 4 °C. The wells were rinsed 3 times, and 0.5% BSA in PBS containing the peroxidase-conjugated goat anti-mouse IgG1 antibody (Catalog #AP124P, Sigma Aldrich; 1:500) was added. The samples were then incubated for 2 h at rt. After rinsing, the substrate solution [phosphate-citrate buffer pH 5.0, o-phenylenediamine (Sigma Aldrich), and H_2_O_2_ (J.T.Baker)] were added, and the samples were incubated for 15 min at rt while protected from light. Finally, 0.1 M H_2_SO_4_ (J.T.Baker) was used to stop the reaction. Absorbances at 492 nm were determined. The cut-off point was defined as double the average absorbance value of the negative controls, and absorbances greater than the cut-off point were interpreted as positive.

### 2.18. Evaluation of Exosome Capacity for Viral Transmission by Lytic Plaque Assay

Exosome samples in serial 10-fold dilutions were prepared in non-supplemented DMEM. Lytic plaque assays were performed as described in Section 2.3.

### 2.19. Exosome Staining with Calcein AM

Paramagnetic-bead-coupled exosomes were suspended in 1 µM Calcein red-orange AM solution (Catalog # C34851, Invitrogen) and incubated for 20 min at 37 °C, protected from light. Fluorescence detection was performed in the FACSCalibur flow cytometer.

### 2.20. Quantification of Exosome Proteins by the Micro-BCA Assay

Protein quantification was performed according to the specifications of the Micro-BCA Protein Assay kit (Thermo Fisher Scientific). The calibration curve was prepared in serial 1:2 dilutions from 2 mg/mL BSA in PBS 1× solution. Exosome samples (150 µL) were added to a flat-bottomed 96-well microtiter plate. Next, the reagent (150 µL) was added and the samples were incubated at 37 °C for 2 h. Absorbances at 562 nm were determined.

### 2.21. Evaluation of the Exosome–Naïve-Vero-Cell Interaction by IF

Calcein AM-stained pmag-bead-coupled exosomes were mechanically uncoupled as described in Section 2.15. Confluent Vero cell monolayers that had been seeded in an 8-well separation chamber slide were stimulated with 1.0 mg of total protein from Calcein AM-stained exosomes, maintained in DMEM with 5% FBS-dEV, and incubated at 37 °C with 5% CO_2_ for 6 h. Cell fixation, blocking, and ZIKV detection were performed as described in Section 2.6. The Olympus IX71 inverted microscope was used for analysis.

### 2.22. Evaluation of the Exosome–Naïve Monocyte Interaction by FACS

Monocytes (1.0 × 10^6^) were stimulated with 1.0 mg of total protein from Calcein AM-stained exosomes, maintained in RPMI-1640 with 5% FBS-dEV, and incubated for 2, 6, 12, and 24 h at 37 °C in 5% CO_2_. Cells were fixed with 2% PFA for 5 min at 4 °C. Fluorescence detection was performed in the FACSCalibur flow cytometer.

### 2.23. Exosome- Stimulation Assays with Naïve Mø and Detection of Virus and Cell Markers by FACS

Monocytes (1.0 × 10^6^ cells) were stimulated with 1.0 mg of total exosome protein, maintained in a non-supplemented RPMI-1640 medium, and incubated for 2 h at 37 °C with 5% CO_2_. Then, 5% FBS-dEV-supplemented medium was added, and the samples were incubated for 96 h at 37 °C with 5% CO_2_. Cells were collected by scraping and pipetting homogenization in sterile microcentrifuge tubes and treated as described in Section 2.9.

### 2.24. Statistical Analysis

All values are presented as the mean ± standard deviation. Statistical analysis was conducted using GraphPad Prism software version 9.5.0 (GraphPad Software Inc., San Diego, CA, USA). Multiple comparisons were assessed by one-way ANOVA with previous verification of data normality by the Kolmogorov-Smirnov test. The Dunnett T3 test was performed to compare treatments vs. control. Statistical significance is denoted as follows: * when *p* < 0.05, ** when *p* < 0.01, and *** when *p* < 0.0001.

## 3. Results

### 3.1. ZIKV Infection Induces the Differentiation and Activation of Human Monocytes 

We determined the best conditions for monocyte (Mø) infection using the purified ZIKV stock. We found that Mø exposed to ZIKV MOI 1, 3, or 5 present the viral E protein on their membranes and that viral mRNA transcripts can be detected 24 h p.i. (Appendix A), indicating active infection. We selected MOI 1 as the infective dose, and the Mø were incubated for 24, 48, 72, and 96 h post-infection (p.i.) to quantify the viral E protein and non-structural protein 1 (NS1) in cell membranes (Figure 1).

The levels of E protein increased significantly (*p* < 0.0001) over time in infected cells compared to uninfected cells (Control Mø). We detected the highest levels at 96 h p.i., when 86.2 ± 3.4% of cells were positive (Figure 1A,B). In the same way, the detected levels of the NS1 protein increased significantly (*p* < 0.0001) during the evaluation period in infected cells compared to Control Mø. The maximum detected levels were reached at 96 h p.i., when 70.9 ± 2.3% of cells were positive for the NS1 protein (Figure 1C,D). The mean fluorescence intensity (MFI) values are shown in Appendix A.

The levels of the E protein were approximately 1.2-fold higher than those of NS1 protein at 96 h p.i. None of the viral proteins was detected in Control cells. The presence of the E protein in the cell membranes may indicate virion adherence before internalization or the presence of new viral progeny being released. Detection of the NS1 protein in the cell membranes is also indicative of active replication, as we used a purified stock. The presence of NS1 on the surface of ZIKV-infected cells has been described previously; however, its role in Zika pathogenesis is not well established, unlike that of the extracellularly secreted form, which is known to be an important virulence factor and which, like the E protein, is one of the major targets for neutralizing and protective antibodies [37,38].

We then evaluated the Mø response to ZIKV infection by detecting different membrane markers of cell differentiation and activation, as those are processes during which exosomes could be released (Figure 2 and Appendix A).

We evaluated the CD14 and CD16 profiles (Figure 2A,B and Appendix A) and found that Control Mø expressed basally high CD14 levels (85.4 ± 5.1%) but very low CD16 levels (1.8 ± 0.5%). Thus, in our model, the classical monocyte phenotype predominated. After ZIKV infection, we detected a significant increase in the percentage values of Mø that were CD14+ (*p* < 0.05), with the highest levels at 96 h p.i. (95.1 ± 2.0%). Similarly, we observed an upward trend in CD16 levels up to 96 h p.i., when 12.9 ± 1.0% of Mø were positive. The increase in the percentage of Mø that were CD16+ was significant (*p* < 0.01) from 24 h p.i. The percentages of ZIKV-infected Mø positive for CD14 and CD16 at 96 h p.i. increased by 1.1- and 7.0-high fold, respectively, compared with baseline. Our findings suggest that the ZIKV infection promotes Mø differentiation from the classical to the intermediate phenotype.

Intermediate Mø are important mediators during acute inflammation [39], so we evaluated multiple antibodies (CD11b, CD11c, CD80, and CD86) in a panel to quantify their degrees of activation (Figure 2C–F and Appendix A).

We determined that uninfected Mø had very low CD11b levels (2.1 ± 0.2%) and that this value increases during ZIKV infection up to 96 h p.i. (90.1 ± 3.4%) (Figure 2C and Appendix A). The percentages of Mø positive for CD11b were significantly (*p* < 0.0001) higher after 24 h p.i. At 96 h, the CD11b levels were 43.3-fold higher compared with the Control Mø. Likewise, we found that Control Mø had low CD11c levels (6.4 ± 0.6%), which also increased during viral infection up to 96 h p.i. (96.7 ± 2.3%) (Figure 2D and Appendix A). The percentages of Mø positive for CD11c were significantly different from the control (*p* < 0.01) after 24 h p.i. At 96 h, the CD11c levels increased by 15.2 times compared with the Control Mø. These results indicate that ZIKV-infected monocytes are activated, expressing a pro-adherent phenotype potentially polarized to a proinflammatory state [40].

In infectious processes, intermediate Mø express costimulatory molecules (CD80 and CD86) that give them the ability to act as mediators of the cellular immune response through antigenic presentation [41]. We detected moderate CD80 levels in Control Mø (28.3 ± 5.3%). Following ZIKV infection, we detected increasing CD80 levels (*p* < 0.0001) up to 96 h p.i. (87.7 ± 4.1%) (Figure 2E and Appendix A). The number of positive cells was 3.1-fold higher compared with Control Mø. In parallel, we found that low CD86 levels were detected in the uninfected Mø (13.2 ± 2.7%). The maximum level detected was observed at 24 h p.i. (60.0 ± 2.8%; *p* < 0.0001), with a sustained decrease after 48 h p.i., indicating that this molecule could be an earlier activation marker (Figure 2F and Appendix A). At 24 h p.i., CD86 levels had increased 4.6 times compared with uninfected cells. The MFI values are shown in Appendix A.

Our findings suggest that, during ZIKV infection, human monocytes differentiate from the classical to the intermediate phenotype and that they enter an activated state that is maintained up to 96 h p.i., with detectable levels of CD11b, CD11c, CD80, CD86, and viral E/NS1 proteins.

### 3.2. Activated Intermediate Monocytes Translocate Endosomal-Trafficking-Associated Proteins and Release Exosomes

We evaluated the levels of tetraspanins (CD63/CD81) and ESCRT-associated protein (TSG101/Alix) in the membranes of ZIKV-infected Mø to elucidate their behavior at 96 h p.i. and its possible relationship with exosome release (Figure 3 and Appendix A).

The tetraspanin CD63 regulates intracellular protein trafficking, and due to its abundance in cellular membranes, it is considered one of the main exosome markers [42,43]. We found that CD63 is present on Mø membranes at baseline (98.7 ± 0.9%; MFI = 802.4 ± 6.2) and that during ZIKV infection, the percentage of positive cells remained stable (>90%; Figure 3A and Appendix A). However, a pattern of decreasing MFI values was observed (Figure 3B). The MFI value at 96 h p.i. was 2.6 times lower than that of uninfected monocytes (*p* < 0.0001), equivalent to a 38.9% reduction compared with the baseline. Our findings demonstrate that in ZIKV-infected Mø, CD63 internalization occurs; this process could be involved in exosome biogenesis, as previously described [44].

The tetraspanin CD81 participates mainly in cell adhesion, and because of its abundance in the plasma membrane, it contributes to early endosome formation, so it is also an exosome marker [45]. In Control Mø, we found a 99.0% ± 0.8% (MFI = 582.1 ± 12.0) positivity rate for CD81, with no changes observed in the ZIKV-infected Mø; however, MFI values increased from 24 h p.i. (MFI = 664.3 ± 17.0), reaching a maximum at 48 h p.i. (MFI = 822.7 ± 11.3, *p* < 0.0001) and remaining stable at 72–96 h p.i. (Figure 3C,D and Appendix A). These findings suggest that more CD81 molecules are expressed on the membranes of ZIKV-infected monocyte (~1.4-fold increase); this change could explain the formation of syncytia (refer to Appendix A) and thus the efficient cell-to-cell ZIKV infection indicated by the high percentages of cells positive for the viral E and NS1 proteins (refer to Figure 1).

TSG101 and Alix are also vesicular-traffic regulators associated with ESCRT, and their detection in exosomes isolates demonstrates their endosomal origin [46]. As expected, we found that uninfected monocytes do not basally express TSG101 (1.3 ± 0.3%; MFI = 10.9 ± 2.0) or Alix (2.2 ± 0.3%; MFI = 12.1 ± 1.9) on their membranes (Figure 3E–H and Appendix A). After ZIKV infection, we detected the presence of TSG101 at the cell membrane from 24 h p.i. The maximum level was observed at 48 h p.i. (79.8 ± 3.2%; MFI = 23.9 ± 2.3, *p* < 0.0001), having increased by 63.4 and 2.2 times in percentage and MFI values, respectively, compared with those of the Control Mø. In parallel, we observed that Alix levels increased significantly from 24 h p.i. (89.7 ± 2.7%; MFI = 27.7 ± 1.9; *p* < 0.0001) and that the percentage rate and MFI values were 41- and 2.3-fold higher, respectively, compared with those of the Control Mø. The presence of TSG101 and Alix, complexed with the NS3 protein, on the membranes of cells infected with some flaviviruses has been found to be necessary for viral release [47,48,49], while TSG101 translocation is also a repair mechanism for the plasma membrane, which could be damaged as a result of infection [50]. These results are consistent with active ZIKV replication in Mø in our model.

Our findings show that ZIKV-infected monocytes display a dynamic of internalization (CD63) and translocation to the plasma membrane (CD81, TSG101, and Alix) of proteins with vesicular-trafficking functions starting at 24 h p.i. This pattern could be related to viral replication and exosome biogenesis. Considering the levels of cell differentiation and activation (refer to Figure 2), we established a time range of 72–96 h p.i. for exosome isolation, characterization, and identification.

Exosomes (Mø exo Control and Mø exo ZIKV) were isolated by differential ultracentrifugation from culture media containing 2.0 × 10^7^ uninfected monocytes and ZIKV-infected monocytes, respectively (Figure 4).

From the NTA data, we found that heterogeneous populations of nanoparticles <200 nm are released, a result consistent with exosome sizes. In exosomes from Control cell (Mø exo Control) isolates, we identified a concentration of 4.8 × 10^9^ ± 2.7 × 10^8^ particles/mL, with an average size of 139 ± 33 nm (Figure 4A), and in Mø exo ZIKV isolates, we found concentrations of 4.1 × 10^10^ ± 1.1 × 10^9^ particles/mL, with an average size of 152 ± 55 nm (Figure 4B). This result indicates that during ZIKV infection, the amount of nanoparticles released is significantly increased (*p* < 0.001) and the size of these particles becomes 1.1-fold greater. In Mø exo ZIKV isolates, we identified a peak around 70 nm that corresponds to a concentration of 5.0 × 10^6^ particles/mL, 0.01% of the total population. This could represent exosomes mixed with viral particles (Appendix A: NTA of purified ZIKV stock). We characterized exosome isolates morphologically by TEM and found heterogeneous populations of cup-shaped extracellular vesicles with well-defined lipid bilayers and sizes <200 nm (Figure 4C).

The endosomal origin of the EV isolates was confirmed performed by detection of CD63, CD81, TSG101, and Alix before coupling with the pmag beads (Appendix A). We found that 51.0 ± 2.7% of pmag-bead-coupled Mø exo ZIKV and the 20.9 ± 3.2% of the pmag-bead-coupled Mø exo Control were CD63+ (Figure 4D and Appendix A). For CD81, 69.6 ± 4.0% of pmag-bead-coupled Mø exo ZIKV and 57.1 ± 5.1% of Control isolates were positive (Figure 4E and Appendix A). The percentages of Mø exo ZIKV CD63+ and CD81+ were increased (2.4- and 1.2-fold higher, respectively) compared with the Control, suggesting high membrane enrichment.

Pmag-bead-coupled exosomes were fixed and permeabilized with 96% methanol for TSG101 and Alix detection. We found that 40.0 ± 3.3% of pmag-bead-coupled Mø exo ZIKV and 28.1 ± 2.9% of pmag-bead-coupled Mø exo Control were TSG101+ (Figure 4F and Appendix A), whereas 30.8 ± 5.0% of pmag-bead-coupled Mø exo ZIKV and 12.5 ± 2.8% of pmag-bead-coupled Mø exo Control were Alix+ (Figure 4G and Appendix A). The percentages of Mø exo ZIKV that were TSG101+ and Alix+ were greater than the corresponding values for the Control (1.4 and 2.5 times, respectively), a result also suggestive of endosomal protein enrichment. The MFI values are shown in Appendix A.

These findings demonstrate that ZIKV-infected monocytes release abundant exosomes that are 1.1-fold larger and enriched with CD63, CD81, TSG101, and Alix compared with exosomes released basally under normal conditions.

### 3.3. ZIKV-Infected Monocyte Exosomes Carry Viral Elements

We analyzed the Mø exo ZIKV TEM images and observed compact, electron-dense structures inside that were not found in the Control images. These structures were suggestive of the presence of virus-like particles inside the exosomes (Figure 5A), so we wondered whether the exosomes might contain viral elements and thus play a role in viral transmission.

Next, we evaluated whether the viral E and NS1 proteins could be detected in the pmag-bead-coupled Mø exo ZIKV isolates. In our previous work, we demonstrated that ZIKV does not cross-react with the anti-CD63-coated paramagnetic beads [26]. Therefore, any viral components detected would be part of the exosome membranes. We found that 29.1 ± 3.3% and 13.4 ± 2.6% of pmag-bead-coupled Mø exo ZIKV were positive for ZIKV E and NS1 protein, respectively (Figure 5B,C and Appendix A); these proteins were not detected in the Control isolates and the samples evaluated as isotype control. These findings indicate that exosomes from ZIKV-infected monocytes can carry viral elements. The MFI values are shown in Appendix A.

According to the results of the NTA, wherein we identified a peak of exosomes mixed with viral particles (refer to Figure 4B), we included an additional step in our experimental design: Mø exo ZIKV isolates were treated to reduce the number of ZIKV virions and purify exosomes (refer to Appendix A) before the exosome function was evaluated in our in vitro model. The Mø exo ZIKV isolates treated with UV and RNase A were identified as Mø exo ZIKV (RNase A + UV), while those isolates purified by mechanical uncoupling from the pmag beads were identified as Mø exo ZIKVp. To verify the performance of the treatments, we tested for free viral RNA in the Mø exo ZIKV (RNase A + UV) and Mø exo ZIKVp isolates through agarose gel electrophoresis (Figure 5D). A visible band of approximately 11 kb, comparable to that in the positive control, was observed in the untreated Mø exo ZIKV isolates, indicating that during differential ultracentrifugation, the viral RNA precipitates. This band was not observed in the Mø exo ZIKVp isolates, and a degraded pattern was detected in the Mø exo ZIKV (RNase A + UV) isolates. These results demonstrate that the treatments eliminate free viral RNA.

We evaluated (by NTA) the degree to which the treated exosomes’ quantity and size were preserved to identify whether significant losses occur that could compromise their functionality in viral transmission. In the Mø exo ZIKV (RNase A + UV) isolates, a concentration of 2.1 × 10^9^ ± 2.0 × 10^8^ particles/mL, with a mean size of 153.3 ± 46.3 nm, was detected (Figure 5E). We observed a significant decrease in the EV population mixed with viral particles compared with the population identified in the Mø exo ZIKV isolates (Figure 4B). We did not identify any change in average size, which suggests the maintenance of structural integrity. For the Mø exo ZIKVp isolates, a concentration of 2.6 × 10^8^ ± 2.2 × 10^7^ particles/mL with a mean size of 133.5 ± 36.7 nm was observed (Figure 5F). For these isolates, we did not find a peak compatible with exosome populations mixed with ZIKV virions. These results suggest that treatment with RNase A and UV radiation, as well as mechanical uncoupling from paramagnetic beads, efficiently reduces the amounts of infectious virions or genomic RNA that may interfere with the functionality assays.

Having demonstrated the preservation of treated exosomes, we confirmed the presence of viral E and NS1 proteins in the Mø exo ZIKV (RNase A + UV) and Mø exo ZIKVp isolates by ELISA (Figure 5G,H). For the E protein, we used the ZIKV stock samples as a positive control (OD_492nm_ = 0.96 ± 0.10), representing 100% positivity. The OD_492nm_ values for the Mø exo ZIKV (RNase A + UV) and Mø exo ZIKVp isolates were 0.33 ± 0.04 and 0.50 ± 0.04, corresponding to positivity rates of 34.6% and 51.8%, respectively. We obtained a positivity rate of 97.1% for the untreated Mø exo ZIKV isolates, which corresponds to the E protein detection from ZIKV virions and the EV population, while in the Mø exo Control and Non-EV SNT (last centrifugation supernatant from the Mø exo ZIKV isolation) samples, the E protein was not detected (Figure 5G).

As expected, the NS1 protein was not detected in the ZIKV stock samples, so we used ZIKV-infected Mø lysates as positive controls (OD_492nm_ = 0.86 ± 0.07; 100% of positivity). We found positivity rates of 28.8% (OD_492nm_ = 0.25 ± 0.05) and 34.7% (OD_492nm_ = 0.30 ± 0.04) for Mø exo ZIKV (RNase A + UV) and Mø exo ZIKVp samples, respectively. For the untreated Mø exo ZIKV isolates, the positivity rate was 67.3%, while the NS1 protein was not detected in the Mø exo Control and Non-EV SNT samples (Figure 5H). The positivity percentages for the E and NS1 protein in the exosome-treated samples were statistically significant (*p* < 0.0001) when compared with the negative control.

The results above demonstrate that exosomes from ZIKV-infected monocytes carry viral E and NS1 proteins. These findings are relevant because the purification treatments reduced the levels of viral components without structural damage to the exosomes; therefore, these Mø exo ZIKV could function as vehicles transferring viral components that favor ZIKV transmission and infection.

In parallel, we evaluated the expression of ZIKV genomic RNA transcripts, specifically of the conserved region that codes for the E protein, in the treated Mø exo ZIKV isolates. For this purpose, the samples were lysed under highly denaturing conditions that induced membrane rupture to release RNA for detection by RT-PCR (Figure 5I). We identified intense visible bands of the 364 bp amplicon from the Mø exo ZIKV isolates and the ZIKV stock (positive control) samples, moderately visible bands from the Mø exo ZIKVp isolates, and slightly visible bands from the Mø exo ZIKV (RNase A + UV) samples. No visible bands were identified from the Mø exo Control and Non-EV SNT samples. These findings show that exosomes from ZIKV-infected monocytes carry viral genomic RNA as part of their cargo.

Therefore, our results indicate that exosomes from ZIKV-infected monocytes carry viral components such as the viral E and NS1 proteins, as well as viral genomic RNA. Along with the suggestive images of virus-like particles inside the exosomes, these results allow us to postulate that the exosomes serve as ZIKV vehicles.

### 3.4. Exosomes from ZIKV-Infected Monocytes Interact with Naïve Cells

Given the possibility that exosomes from ZIKV-infected monocytes carry viral elements, we evaluated their potential to promote viral transmission and infection through interaction with naïve cells (Figure 6).

We performed lytic plaque assays on confluent Vero cell monolayers. Cells were inoculated with serial dilutions of the different treatments of the Mø exo ZIKV and Mø exo Control samples (Figure 6A). Plaque formation was observed on day 5 post-stimulus with Mø exo ZIKV and ZIKV (MOI 1) and from day 8 post-stimulus with Mø exo ZIKV (RNase A + UV) and Mø exo ZIKVp. Lytic plaques were detected up to dilutions of 10^−7^ for untreated Mø exo ZIKV, 10^−2^ for Mø exo ZIKV (RNase A + UV), and 10^−5^ for Mø exo ZIKVp, indicating transmitted infectious ZIKV titers of 3.7 × 10^7^ ± 1.9 × 10^7^ PFU/mL, 1.4 × 10^3^ ± 4.9 × 10^2^ PFU/mL, and 1.6 × 10^6^ ± 7.1 × 10^5^ PFU/mL, respectively. For the Mø exo Control and Non-EV SNT samples, no lytic plaques were observed (Figure 6B). These findings demonstrate the ability of purified exosomes to transmit infectious viral particles. The infection (identified by lytic plaque formation after UV radiation, RNase A assays, or pmag-bead-uncoupling treatments) suggests that exosomes protect the infective components from degradation by physical or chemical treatments.

To demonstrate that exosomes, by transferring viral RNA or complete viral particles, facilitate viral transmission through their interaction with naïve cells, we performed stimulation assays wherein exosome isolates were stained with Calcein AM. Calcein AM is a non-fluorescent compound that forms a fluorescent anion when it is hydrolyzed by intracellular esterases. As it is known that exosomes contain esterases, this stain can be used to detect for intact EV [51]. Hence, we evaluated the Calcein AM labeling of the pmag-bead-coupled Mø exo ZIKV and Mø exo Control to verify that exosomes can be detected by FACS (Figure 6C and Appendix A). We found that 31.2 ± 3.1% (*p* < 0.0001) of pmag-bead-coupled Mø exo ZIKV fluoresced, while 22.0 ± 2.6% (*p* < 0.0001) of the coupled Mø exo Control fluoresced. No fluorescence emission was identified from uncoupled paramagnetic beads stained with Calcein AM, indicating that non-specific fluorescence did not occur. The MFI values are shown in Appendix A.

In parallel, we found that the Mø exo ZIKVp and Mø exo Control interacted with naïve Vero cells. We observed the presence of viral particles in the Mø exo ZIKVp isolates, which confirmed that the exosomes from ZIKV-infected monocytes carry viral elements, making them efficient vehicles for viral transmission (Figure 6D).

Exosomes can interact with recipient cells through multiple pathways. These interactions allow exosomes to mediate intercellular communication and induce changes in cell phenotype [18,19]. To evaluate the interaction between exosomes and naïve Mø, we performed stimulation assays using uncoupled Calcein AM-labeled exosomes on naïve (unlabeled) cells and then quantified their fluorescence emission by FACS (Figure 6E,F and Appendix A). We found that the maximum fluorescence peak, associated with the interaction between Calcein AM-labeled exosomes and naïve Mø, occurred 6 h post-stimulation in the cells stimulated with Mø exo ZIKVp (20.2 ± 2.9%; *p* < 0.0001), as well as in those stimulated by the Mø exo Control (14.5 ± 1.3%; *p* < 0.0001); however, fluorescence was detected from 2 h post- stimulation. After 6 h, the MFI values decreased, and at 24 h, the values were similar to those detected at 2 h, suggesting that internalization potentially occurred. The MFI values are shown in Appendix A.

These findings demonstrate that interaction between exosomes and naïve cells occurs and that exosome internalization may favor viral transmission and infection.

### 3.5. Exosome Interaction Favors Infection, Differentiation, and Activation of Naïve Monocytes

After we demonstrated the interaction between exosomes and naïve monocytes, we evaluated the viral infection of these cells by quantifying the ZIKV E and NS1 protein levels after incubation for 96 h (Figure 7).

We found that naïve monocytes stimulated with Mø exo ZIKVp and Mø exo ZIKV (RNase + UV) present significant E protein levels (*p* < 0.0001), with this protein detected in 68.4 ± 3.2% and 58.7 ± 4.3% of cells, respectively (Figure 7A,B). The E protein was not detected in monocytes stimulated with Mø exo Control (2.1 ± 0.2%) and Non-EV SNT (2.9 ± 0.2%). In untreated Mø exo ZIKV-stimulated monocytes or ZIKV-infected Mø, which were used as positive controls, the E protein was detected in more than 70% of the cells.

We also found that Mø exo ZIKVp- and Mø exo ZIKV (RNase + UV)-stimulated monocytes presented significantly high levels of the NS1 protein (*p* < 0.0001), with this protein detected in 49.6 ± 7.4% and 31.6 ± 6.3% of cells, respectively. The NS1 protein was not detected in cells stimulated with Mø exo Control (1.6% ± 0.4%) and Non-EV SNT (3.8% ± 0.8%). In the positive controls, this protein was detected in more than 61% of the monocytes (Figure 7C,D). The MFI values are shown in Appendix A.

Our results suggest that stimulation with treated Mø exo ZIKV promotes active infection in naïve monocytes, as demonstrated by the high levels of the viral E and NS1 proteins. Therefore, exosomes from ZIKV-infected monocytes are an efficient alternative mechanism for viral transmission, acting as Trojan horses that facilitate viral infection in a receptor-independent manner.

Finally, we evaluated how the naïve monocytes respond to the different exosome isolates stimulation, using detection of the markers CD14, CD16, CD11b, CD11c, CD80, and CD86 to determine the degrees of differentiation and activation (Figure 8).

We found that 96.3 ± 3.6% and 95.5 ± 3.2% of the Mø exo ZIKVp- and Mø exo ZIKV (RNase A + UV)-stimulated cells, respectively, were CD14-positive (Figure 8A and Appendix A), while 10.6 ± 0.5% of the Mø exo ZIKVp-stimulated monocytes and 8.4 ± 0.6% of the Mø exo ZIKV (RNase A + UV)-stimulated cells were CD16-positive (Figure 8B and Appendix A). The rate of positivity for CD14 was 1.1-fold higher after Mø exo ZIKVp stimulation (*p* < 0.01). CD16 levels increased 3.7- and 2.9-fold after Mø exo ZIKVp and Mø exo ZIKV (RNase A + UV) stimulation (*p* < 0.0001 and *p* < 0.01, respectively) compared with control cells. In terms of MFI values, we found increases of 2.2- and 2.0-fold for Mø exo ZIKVp- and Mø exo ZIKV (RNase A + UV)-stimulated cells, indicating that the amount of CD16 increased on the membranes of stimulated cells compared to those of unstimulated cells (The MFI values are shown in Appendix A). The positivity rates in the untreated Mø exo ZIKV isolate-stimulated and ZIKV (MOI 1)-infected cells, which were used as positive controls, were >97% for CD14 and >13% for CD16. These results demonstrate that stimulations with exosomes from ZIKV-infected monocytes promotes cell differentiation from the classical to the intermediate phenotype.

Exosome interactions also induce cell activation (Figure 8C,D and Appendix A). We found that 77.4 ± 2.2% and 76.2 ± 2.5% of the Mø exo ZIKVp- and Mø exo ZIKV (RNase A + UV)-stimulated naïve monocytes were CD11b-positive, respectively, while 37.9 ± 1.3% and 36.8 ± 1.0% were CD11c-positive. The observed rates of CD11b and CD11c positivity were >25 and >10.2 times higher (*p* < 0.0001), respectively, compared with Control Mø. For the positive controls, CD11b was detected in >94% and CD11c in >77% of cells.

Also, we found that 69.4 ± 1.6% and 47.9 ± 1.9% of Mø exo ZIKVp- and Mø exo ZIKV (RNase A + UV)-stimulated monocytes were CD80+, respectively, while 49.5 ± 1.6% and 22.6 ± 1.8% (MFI = 20.5 ± 1.5) were CD86+ (Figure 8E,F and Appendix A). For cells exposed to Mø exo ZIKVp stimuli, the CD80 levels were 5.5 times higher (*p* < 0.0001) and CD86 levels were 3.2-fold higher (*p* < 0.0001) compared with control cells. For Mø exo ZIKV (RNase A + UV) stimuli, the CD80 and CD86 levels were 3.5 and 1.4 times higher, respectively (*p* < 0.0001) compared with control cells. Among the positive controls, >77% were CD80+, and >50% CD86+.

These results suggest that stimulation with exosomes from ZIKV-infected monocytes containing viral components also induce the differentiation and activation of naïve cells. Given that they can also facilitate infection, these exosomes may promote a long infection–activation–infection cycle with constant exosome release potentially contributing to an inflammatory state, wherein ZIKV is “silently” transmitted between the host cells. This mechanism also should be considered a relevant form of non-vectorial transmission in the pathogenesis of Zika fever.

## 4. Discussion

The incidence of diseases caused by arboviruses continues to rise worldwide, and these diseases have expanded to new areas. Arboviruses represent a significant public health challenge because of the high rates of severe clinical outcomes and the continuous potential to cause epidemic outbreaks. Mosquito-borne flaviviruses have spread rapidly due to climate change and are characterized by highly virulent circulating strains, urban-cycle-adapted vectors, and susceptible human hosts, of whom up to 400 million are infected annually [52,53,54].

The recent Zika virus epidemic in the Americas (2015–2016), which was declared a Public Health Emergency of International Concern, evidenced the mosquito vectors’ wide distribution and the possibility of non-vectorial viral transmission as part of the pathogenesis of Zika fever [7,8,9,55,56]. Knowledge of the molecular and cellular mechanisms underlying virus-host interactions, including the essential role of EV in the, is important to understanding viral transmission and disease progression, especially for the severe forms mainly characterized by neurological alterations [57]. It is known that EV have a role in facilitating arbovirus transmission between arthropod cells and from arthropod cells to mammalian cells [58]. Hence, they represent a key mechanism of vectorial transmission and disease establishment. Previously, we demonstrated that EV from ZIKV-infected mosquito cells transferred viral components to human monocytes and endothelial vascular cells, facilitating infection and modifying the cells’ behavior, which allowed us to conclude that EV participate in ZIKV vectorial transmission [26]. The role of EV in the flaviviruses’ vectorial transmission has been described for DENV and the Langat virus (LGTV, closely related to tick-borne encephalitis virus) [22,23,24,59].

The concern about non-vectorial transmission, including person-to-person transmission, is that this mechanism could limit ZIKV clearance or promote evasion of the immune response, both effects that are directly involved in severe clinical outcomes of Zika fever. In human hosts, ZIKV has broad cell tropism as part of its replicative cycle; however, monocytes are the primary targets [11,12,13,14,15]. Monocytes’ function is essential for pathogen control and elimination, but dysregulation of their activity may contribute to inflammatory and degenerative diseases. According to published findings, in response to infection, monocytes release humoral factors and EV as differentiation/activation products and thus could be an alternative mechanism for viral transmission. Based on the “Trojan exosome” hypothesis, the replicative cycle of RNA viruses and exosome biogenesis may converge, allowing exosomes to serve as carriers of viral components [14,20,21,60,61]. Therefore, exosomes from ZIKV-infected monocytes may be required for ZIKV transmission and infection within human tissues as part of the non-vectorial viral-transmission pathway.

In this study, we evaluated whether exosomes from ZIKV-infected monocytes participate in ZIKV cell-to-cell transmission through the transfer of viral RNA or through allowing viruses to infect cells via receptor-independent pathways, enhancing viral replicative capability as well as modulating cell responses. In these respects, monocytes would constitute an effective alternative mode of propagation. We found that ZIKV infection induces monocyte differentiation and activation (Figure 1 and Figure 2), including the release of high amounts of exosomes. It is known that intermediate monocyte subpopulations increase in Zika fever patients, and these cells are permissive to infection and contain viral RNA, suggesting that they act as viral reservoirs. In this sense, flaviviruses like ZIKV, DENV, and WNV share the strategy of using monocytes as Trojan horses that allow the viruses to spread to immunologically privileged tissues [11,12,13]. Our results showed that intermediate-activated monocytes, in response to ZIKV infection, exhibited adhesion (CD11b+ CD11c+) properties and expressed co-stimulatory molecules (CD80+ CD86+). This profile may promote early innate immune changes to induce an inflammatory state and contribute to cellular responses in the human host. We also found that monocyte ZIKV infection induced tetraspanin CD63 internalization and translocation of tetraspanin CD81 and ESCRT-associated proteins TSG101 and Alix to the plasma membrane (Figure 3). These changes may be associated with the viral replicative cycle and EV release. We identified a reduction of 38.9% in CD63 levels (MFI values) in ZIKV-infected monocytes at 96 h p.i. This behavior has been described as a cellular signal associated with virions and EV release [44]. It is known that tetraspanins are necessary for the entry and budding of various viruses (e.g., hepatitis C virus, a *Flaviviridae* family member, requires CD81 to infect hepatocytes), and the interactions between TSG101 and Alix and flavivirus (DENV, YFV, and JEV) proteins serve as key signals to promote viral release. Translocation of these cellular proteins is also associated with repair mechanisms in infected cells [47,48,49,50,62,63]. In this study, we identified low levels of CD63 and high levels of CD81, TSG101, and Alix on the membranes of ZIKV-infected monocytes as a molecular profile that could be associated with the convergence of the ZIKV replicative cycle and exosome biogenesis.

As mentioned earlier, ZIKV-infected monocytes can be Trojan horses that allow the virus access to immunologically privileged tissues, and thus EV release can also contribute to ZIKV cell-to-cell transmission and infection, as well as to promoting a proinflammatory state, making EV release as an effective part of the non-vector transmission pathway. In the present work, we isolated exosomes from the culture media of intermediate-activated ZIKV-infected monocytes at 72–96 h p.i. (Figure 4). Differential ultracentrifugation allowed us to obtain large amounts of exosomes. Paramagnetic-bead-based isolation was used for exosome identification, and characterization was carried out by NTA and TEM. ZIKV-infected monocytes release heterogeneous populations of CD63+ CD81+ TSG101+ Alix+ exosomes (Mø exo ZIKV) in concentrations of ~4.1 × 10^10^ particles/mL with an average size of 152 nm. The Mø exo ZIKV were 1.1-fold larger than the exosomes released by uninfected cells, and the detection of tetraspanins and ESCRT-associated proteins demonstrated their endosomal origin. Other studies of exosomes in DENV- and JEV-infected monocytes, macrophages, or dendritic cells have described the release of large amounts of CD63+ exosomes with an average size of 30–120 nm, isolated from culture media at 72–120 h p.i. [25,32,64]. As the function of monocyte exosomes has not previously been described for ZIKV infection, this study represents the first such characterization.

It has been proposed that the size variation in the EV from infected cells could be due to the incorporation of viral components as internal content [23]. We analyzed the Mø exo ZIKV TEM images and observed that the cup-shaped exosomes contained some compact electron-dense structures that were not found in the images of uninfected Mø exosome. Therefore, we evaluated the presence of viral elements at the membranes of Mø exo ZIKV (Figure 5). First, we found that ~29.1% and ~13.4% of paramagnetic-bead-coupled Mø exo ZIKV were positive for ZIKV E and NS1 proteins, respectively. The presence of viral antigens was the first evidence suggesting that Mø exo ZIKV could function as viral vehicles; however, some exosome populations can match in size and density with infective virions [65,66] that co-precipitate during EV isolation. Considering this possibility, we identified by NTA that some Mø exo ZIKV populations were ≤70 nm in size, which could correspond to a virion–EV mixture. Certainly, in the natural infection cycle, mixture of virions and EV populations are released and interact with host cells; however, to assess the role of Mø exo ZIKV in viral transmission, we purified the exosomes by inactivation of ZIKV by UV radiation and RNase A treatment (Mø exo ZIKV [RNase A + UV]) or by uncoupling from anti-CD63-coated paramagnetic beads (Mø exo ZIKVp). We observed that after these treatments, nanoparticles of ~153.3 nm from the Mø exo ZIKV (RNase A + UV) isolates and ~133.5 nm from the Mø exo ZIKVp isolates were recovered without significant traces of nanoparticles smaller than 70 nm, the size of ZIKV viral particles (Appendix A and Figure 5E,F). Treatments with UV radiation and RNase A, as well as mechanical uncoupling from paramagnetic beads, were efficient in reducing the amounts of infectious ZIKV virions or viral RNA, which might otherwise have interfered with the functionality assays. The integrity of the exosomes post-treatment suggests that they resist physical or chemical damage, thereby protecting their internal cargo.

Other studies of EV and flaviviruses have reported that exosomes from infected cells carry viral elements. For ZIKV, RNA transcripts and viral E protein have been demonstrated to occur in exosomes [26,27,28]. For DENV, complete genomes, capsid (C) protein mRNA, and E, NS1, and NS3 proteins have been described [22,25,32]. For JEV and WNV, the NS1 and E protein mRNAs, respectively, have been shown to occur in exosomes [24,67]. In our model, we confirmed the presence of E and NS1 proteins in the membranes of the purified Mø exo ZIKV isolates, as well as the presence of viral RNA E transcripts as part of their internal cargo after membrane rupture. These proteins and RNAs were not detected in the exosomes from control cells. Also, the TEM images suggesting the presence of virus-like particles enclosed in the exosomes indicate that exosomes may have the capacity to transport complete virions, as previously reported for DENV [23], hepatitis C virus [68,69,70], hepatitis A virus [71], LGTV [24], and human immunodeficiency virus type 1 (HIV-1) [72]. For all of the above reasons, we conclude that exosomes from ZIKV-infected monocytes function as vehicles for viral components that may allow evasion of the host’s immune response, limiting viral clearance and favoring dissemination. This mechanism could partly explain how ZIKV reaches and remains in immunologically privileged tissues and triggers severe forms of Zika fever.

As a result of these significant findings, we first evaluated the role of Mø exo ZIKV in viral transmission by the formation of lytic plaques on Vero cells. This model allowed us to demonstrate, quantitatively and visually, that infective virus transmission occurs as a result of exposure to Mø exo ZIKV isolates (Figure 6). Observation of lytic plaques after stimulation with Mø exo ZIKVp and Mø exo ZIKV (RNase + UV) yielded transmitted infectious viruses titers of ~1.6 × 10^6^ and ~1.4 × 10^3^ PFU/mL, respectively, suggesting that exosomes from ZIKV-infected monocytes favor non-human-primate host-cell infection and corroborated the previous finding that exosomes protect infective viral components from degradation by physical or chemical treatments. Subsequently, we wanted to demonstrate that viral transmission is mediated by exosome stimulation through the exosomes’ interaction with naïve cells, which may allow the transfer of viral RNA or of complete virions. Exosomes can interact with, be taken up by, and enter the recipient cells through different mechanisms such as clathrin-, caveolae-, or receptor-dependent internalization; lipid-raft-mediated internalization; micropinocytosis; phagocytosis; and membrane fusion. These interactions allow exosomes to mediate intercellular communication and to induce changes in the phenotypes of acceptor cells [18,19]; however, determining the specific route of this interaction was outside the aims of this study. Here, we demonstrated that interaction between exosomes and naïve cells (Vero and monocytes) occurs, probably by membrane fusion between 2 and 6 h, through Calcein AM-stained exosome-stimulation assays. This interaction may favor viral infection via a receptor-independent mechanism and likely constitutes an efficient mechanism of viral transmission.

The capacity of EV to transmit flaviviruses to naïve cells of the same cell line as those that gave rise to them and to promote active infection has been described for ZIKV in EV models of C6/36 mosquito cells and murine cortical neurons [26,27] and for DENV in C6/36 mosquito-cell models [22,23]. Our results show that stimulation by exosomes from ZIKV-infected monocytes facilitates active viral infection in naïve monocytes (Figure 7). We detected the ZIKV E/NS1 proteins at 96 h post-stimulus. The E protein was detected in ~68.4% and ~58.7% of Mø exo ZIKVp- and Mø exo ZIKV (RNase + UV)-stimulated cells, while the NS1 protein was detected in ~49.6% and ~31.6%, respectively, suggesting that exosomes favor cell-to-cell transmission and promote active infection. Hence, these exosomes are an efficient pathway for viral transmission, acting as Trojan horses. The existence of such Trojan exosomes may have important implications, as their effects could promote “silent” viral replication, both because the rate of viral E/NS1 protein production is comparable to that seen with ZIKV (MOI 1)-mediated infection and because the exosomes may facilitate immune evasion. For severe cases of Zika fever, there are many pending questions regarding how the virus disseminates from its point of entry to new host cells, how it gains access to restricted sites, and how exosomes are also able to transmit cargo over long distances (e.g., through peripheral blood circulation) and interact with different host cells, contributing to ZIKV infection and hindering clearance.

We also found that exosomes from ZIKV-infected monocytes participate in the differentiation and activation of naïve cells (Figure 8), promoting an infection–activation–infection cycle in human monocytes and potentially in any other acceptor cell. This cycle could be characterized by continuous EV production, favoring establishment of a long-lasting inflammatory state. These findings may contribute to our understanding of clinical manifestations like neurologic complications or autoimmunity resulting from post-infectious immune responses or direct viral neurotropism, such as have been reported in severe Zika fever cases.

We recognize that the pathogenesis of Zika fever is complex and depends on the functions of other infected cells and their products in response to infection. This study aimed to evaluate the function of ZIKV-infected monocyte exosomes in cell-to-cell viral transmission. We focused on monocytes because they are distributed in the bloodstream and tissues as sentinels, and in the context of ZIKV and other arbovirus infections, represent the first virus–cell contact [73]. Their function may thus define the clinical outcome (asymptomatic, mild, or severe disease). However, these cells and their differentiation/activation products (extracellular vesicles) are only the beginning of the complex mechanism of Zika pathogenesis, which includes other viral target cells, like endothelial cells, neuronal cells, trophoblasts, etc., with their respective products generated in response to infection.

Further investigation is necessary to establish the impact of Mø exo ZIKV on Zika fever pathogenesis, as ZIKV exploits the exosome pathway for its assembly/budding and for the transfer of viral components. Knowledge of these mechanisms can be used for the development of new therapeutic tools to address severe Zika virus disease. In addition, exosomes from ZIKV-infected monocytes could be utilized as diagnostic or prognostic markers and as vaccine antigens for disease prevention or to halt disease progression.

## 5. Conclusions

Zika virus infection induced differentiation of monocytes from the classical (CD14+ CD16−) to the intermediate (CD14+ CD16+) phenotype. The resulting activation state, which was maintained up to 96 h post-infection, as characterized by detectable levels of viral E and NS1 proteins, as well as by CD11b, CD11c, CD80, and CD86, which are proteins associated with a pro-adherent phenotype and pro-inflammatory state. In the activated intermediate ZIKV-infected monocytes, internalization of one vesicular-trafficking-associated protein (CD63) and translocation of others (CD81, TSG101, and Alix) may be involved in exosome biogenesis and release. We demonstrated that ZIKV-infected monocytes release abundant CD63+ CD81+ TSG101+ Alix+ exosomes < 200 nm in size that carry viral elements like viral E and NS1 proteins and viral genomic RNA transcripts. Interaction between exosomes and naïve cells favors viral transmission and active infection, as demonstrated by the high levels of the viral E and NS1 proteins, and promotes the differentiation and activation of naïve cells.

Our results suggest that ZIKV uses cellular pathways such as exosome biogenesis to efficiently maintain its replicative cycle via receptor-independent mechanisms. These cellular pathways allow the transfer of viral RNA or infectious virions, and they favor evasion of the immune response, which leads to low rates of ZIKV clearance. Stimulation with exosomes from ZIKV-infected monocytes induces an infection–activation–infection cycle that may establish a long-lasting pro-inflammatory state (Figure 9).

Exosomes from ZIKV-infected monocytes are significant because, as a product of the primary cellular target of ZIKV, they may contribute to viral transmission, infection, and activation of other target cells and promote viral infiltration into immune-sheltered tissues (brain, placenta, testes, etc.). They may thus prolong viral clearance by facilitating evasion of the immune response and favor establishment of a pro-inflammatory state. These effects may impact disease progression [11,74].

## Figures and Tables

**Figure 1 cells-13-00144-f001:**
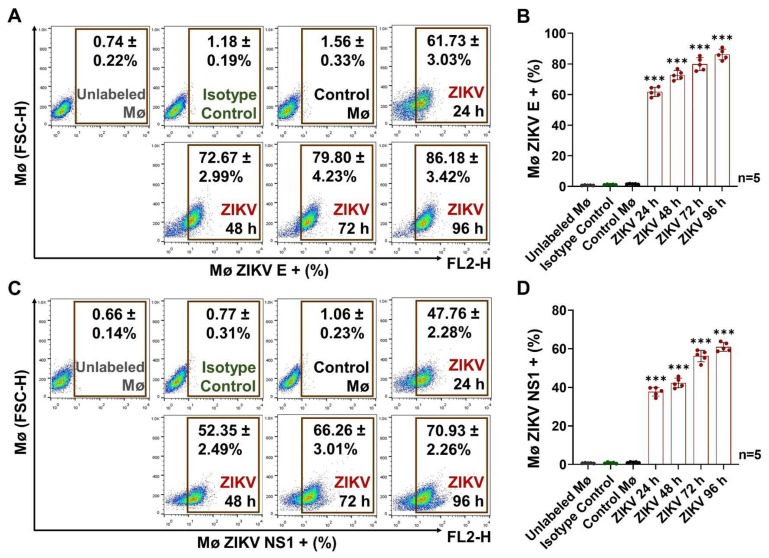
Monocyte (Mø)–ZIKV MOI 1 infection curve. (**A**) Detection of ZIKV E protein at 24, 48, 72, and 96 h p.i. (representative dot plots) by FACS. (**B**) Percentages of Mø positive for protein E. (**C**) Detection of ZIKV NS1 protein at 24, 48, 72, and 96 h p.i. (representative dot plots) by FACS. (**D**) Percentages of Mø positive for protein NS1. The percentages of Mø positive for viral proteins E or NS1 were compared with the Control Mø values by one-way ANOVA. Statistical significance is denoted as *** when *p* < 0.0001. Unlabeled Mø (gray), Isotype Control (green), Control Mø (black), and ZIKV (red).

**Figure 2 cells-13-00144-f002:**
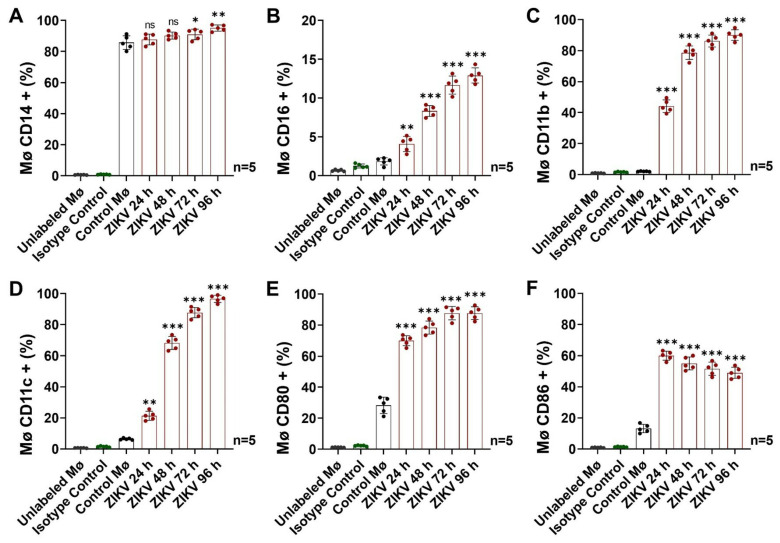
Cell-marker detection (by FACS) at 24, 48, 72, and 96 h p.i. in ZIKV-infected Mø. (**A**) Percentages of Mø positive for CD14. (**B**) Percentages of Mø positive for CD16. (**C**) Percentages of Mø positive for CD11b. (**D**) Percentages of Mø positive for CD11c. (**E**) Percentages of Mø positive for CD80. (**F**) Percentages of Mø positive for CD86. Representative dot plots are shown in Appendix A. The percentages of cells positive for CD14, CD16, CD11b, CD11c, CD80, and CD86 were compared with the Control Mø values by one-way ANOVA. Statistical significance is denoted as follows: * when *p* < 0.05, ** when *p* < 0.01, and *** when *p* < 0.0001. ns = no significance.

**Figure 3 cells-13-00144-f003:**
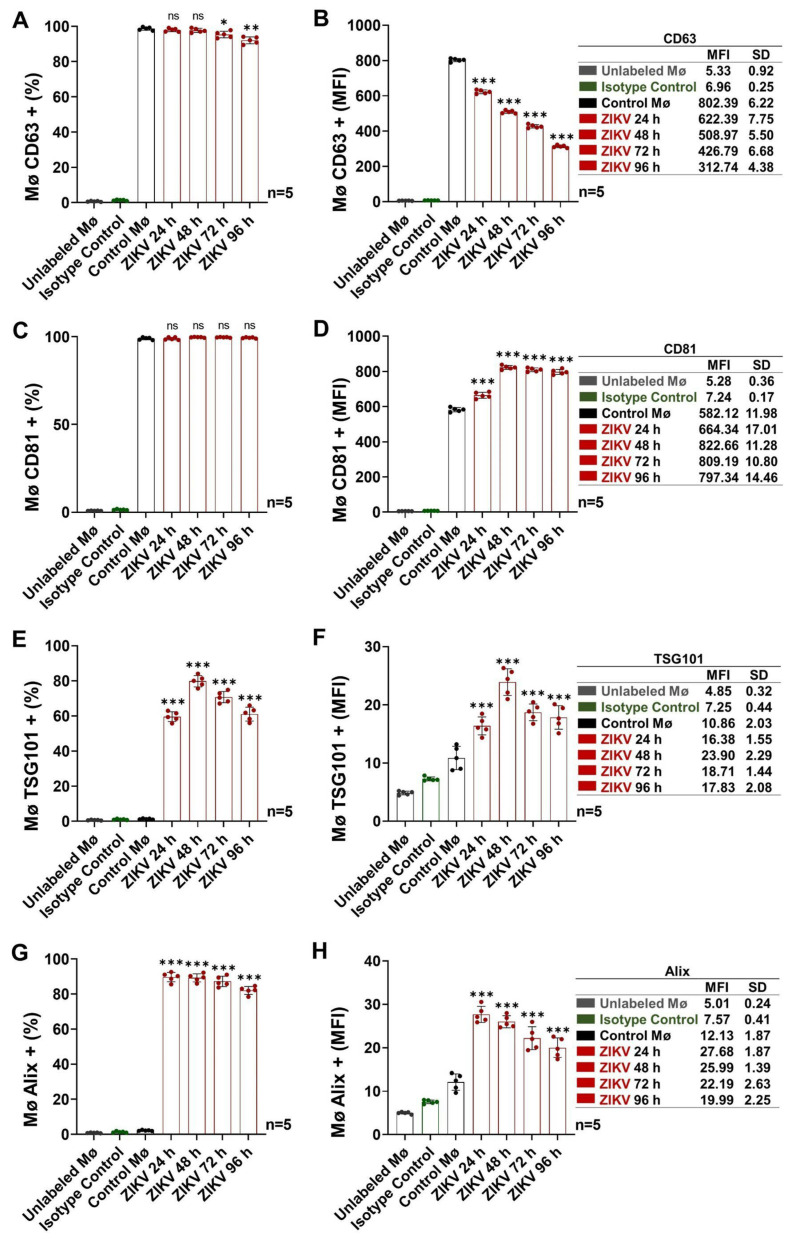
Detection of endosomal-trafficking-associated protein (by FACS) at 24, 48, 72, and 96 h p.i. in ZIKV-infected Mø. (**A**) Mø CD63+ percentages. (**B**) Mø CD63+ MFI values. (**C**) Mø CD81+ percentages. (**D**) Mø CD81+ MFI values. (**E**) Mø TSG101+ percentages. (**F**) Mø TSG101+ MFI values. (**G**) Mø Alix+ percentages. (**H**) Mø Alix+ MFI values. Representative dot plots are shown in Appendix A. The CD63, CD81, TSG101, and Alix levels (percentages and MFI values) were compared with the Control Mø values by one-way ANOVA. Statistical significance is denoted as follows: * when *p* < 0.05, ** when *p* < 0.01, and *** when *p* < 0.0001. ns = no significance.

**Figure 4 cells-13-00144-f004:**
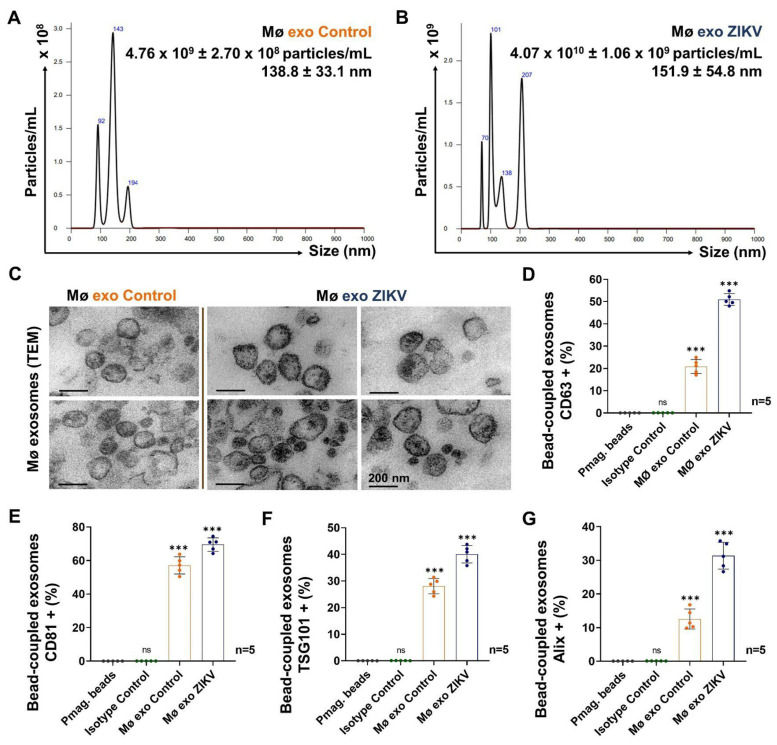
Exosome characterization and identification. (**A**) NTA from the Mø exo Control isolates. (**B**) NTA from the Mø exo ZIKV isolates. Histograms show the representative mean ± SD of the nanoparticles’ concentration (particles/mL) and size (nm) from five independent experiments. (**C**) TEM images from the Mø exo Control and Mø exo ZIKV isolates (200 nm scale). (**D**) Pmag-bead-coupled exosomes CD63+ percentages detected by FACS. (**E**) Pmag-bead-coupled exosomes CD81+ percentages detected by FACS. (**F**) Pmag-bead-coupled exosomes TSG101+ percentages detected by FACS. (**G**) Pmag-bead-coupled exosomes Alix+ percentages detected by FACS. Representative dot plots are shown in Appendix A. The percentages of pmag-bead-coupled exosomes positive for CD63, CD81, TSG101, and Alix were compared with the values for the pmag beads by one-way ANOVA. Statistical significance is denoted as *** when *p* < 0.0001. ns = no significance. Pmag-beads (gray), Isotype control (green), Mø exo Control (orange), and Mø exo ZIKV (dark blue).

**Figure 5 cells-13-00144-f005:**
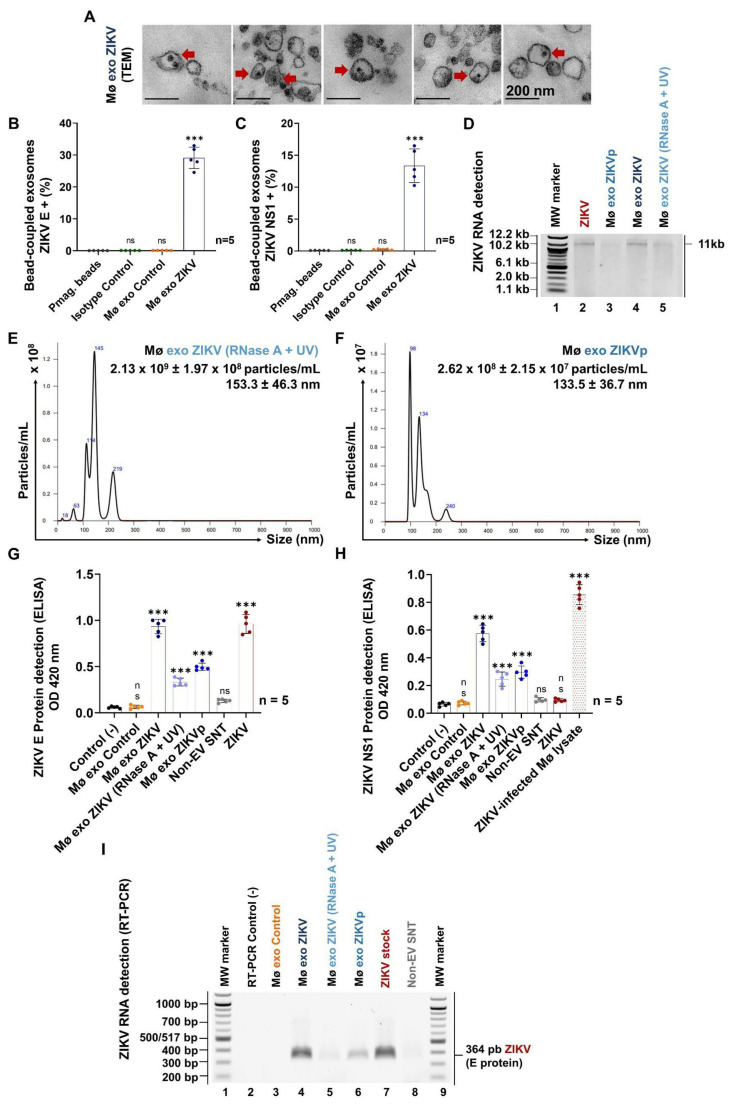
ZIKV-infected Mø exosomes carry viral elements. (**A**) TEM images from the Mø exo ZIKV isolates (200 nm scale). Electrodense bodies, compatible with identification as virus-like particles, are marked with red arrows. (**B**) Percentages of pmag-bead-coupled exosomes positive for ZIKV E protein. (**C**) Percentages of pmag-bead-coupled exosomes positive for ZIKV NS1 protein. (**D**) Detection of ZIKV RNA in Mø exo ZIKV, Mø exo ZIKV (RNase A + UV), and Mø exo ZIKVp isolates. The RNA degradation pattern was visualized on 2% ethidium bromide-stained 1.2% agarose gel. (**E**) NTA from the Mø exo ZIKV (RNase A + UV) isolates. (**F**) NTA from the Mø exo ZIKVp isolates. (**G**) Detection of ZIKV E protein by ELISA. (**H**) Detection of ZIKV NS1 protein by ELISA. (**I**) Detection of ZIKV RNA (by RT-PCR) in the Mø exo ZIKV isolates. The 364 bp amplicon was visualized on 2% ethidium bromide-stained 1.2% agarose gel. Statistical significance is denoted as *** when *p* < 0.0001. ns = no significance. Pmag-beads (gray), Isotype control (green), Control Mø (black), Mø exo Control (orange), Mø exo ZIKVp (sky blue), Mø exo ZIKV (RNase A + UV) (light blue), Mø exo ZIKV (dark blue), Non-EV SNT (light gray), and ZIKV (red).

**Figure 6 cells-13-00144-f006:**
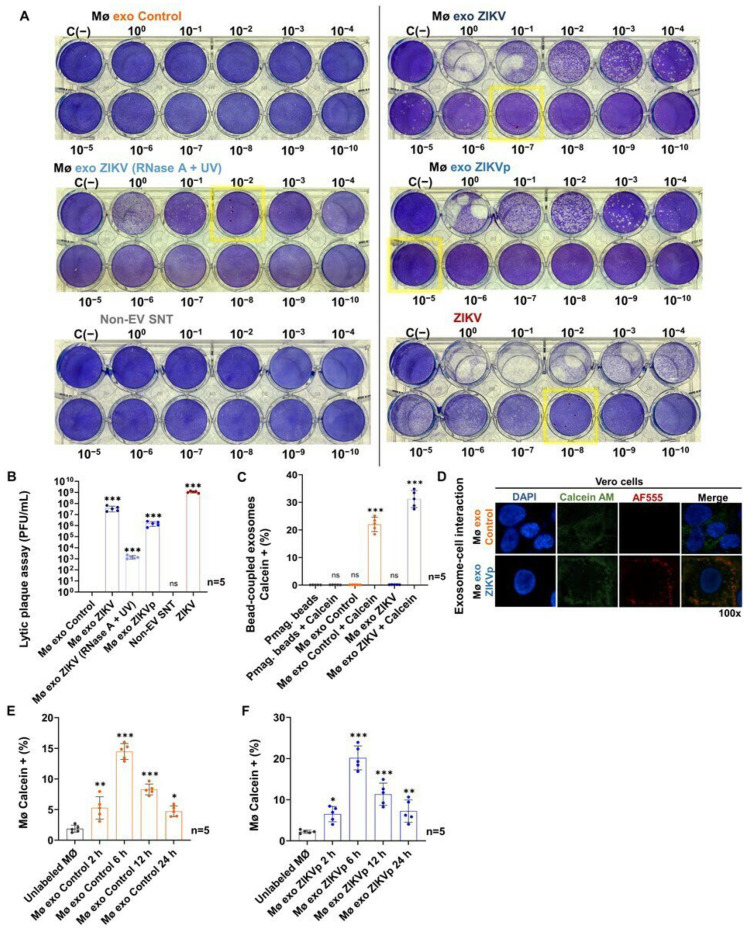
Interactions between exosomes and naïve cells promote ZIKV transmission. (**A**) Representative images of the lytic plaque assays conducted with exosome stimuli and naïve Vero cells. The yellow box indicates the last dilution wherein lytic plaques were counted. (**B**) Titers (PFU/mL) of infective ZIKV transmitted by exosomes. (**C**) Percentages of pmag-bead-coupled exosomes positive for Calcein AM. The representative dot plots are shown in Appendix A. (**D**) Interaction between exosomes and naïve Vero cells detected by fluorescence microscopy (100×) at 6 h p.i. ZIKV E protein (red) and Calcein AM-stained exosomes (green). (**E**) Detection of the interaction between exosomes (Mø exo Control) and naïve Mø at 2, 6, 12, and 24 h p.i. The representative dot plots are shown in Appendix A. (**F**) Detection of the interaction between exosomes (Mø exo ZIKVp) and naïve Mø at 2, 6, 12, and 24 h p.i. The representative dot plots are shown in Appendix A. Statistical significance is denoted as follows: * when *p* < 0.05, ** when *p* < 0.01, and *** when *p* < 0.0001. ns = no significance.

**Figure 7 cells-13-00144-f007:**
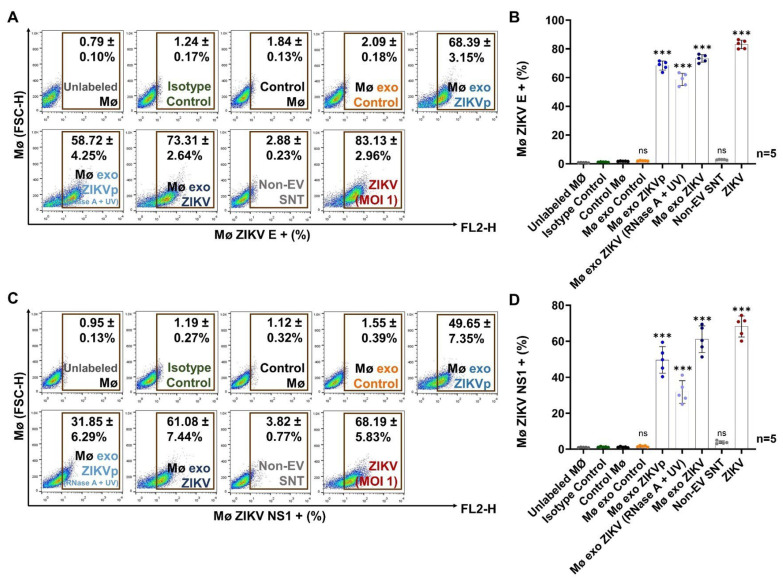
Interaction between ZIKV-infected monocyte exosomes and naïve Mø favors viral infection. (**A**) Detection of ZIKV E protein at 96 h post-stimulus with exosome isolates (representative dot plots) by FACS. (**B**) Percentages of Mø positive for E protein. (**C**) Detection of ZIKV NS1 protein at 96 h post-stimulus with exosome isolates (representative dot plots) by FACS. (**D**) Percentages of Mø positive for NS1. The ZIKV E and NS1 levels in stimulated Mø were compared with the Control values by one-way ANOVA. Statistical significance is denoted as *** when *p* < 0.0001. ns = no significance. Unlabeled Mø (gray), Isotype control (green), Control Mø (black), Mø exo Control (orange), Mø exo ZIKVp (sky blue), Mø exo ZIKV (RNase A + UV) (light blue), Mø exo ZIKV (dark blue), Non-EV SNT (light gray), and ZIKV (red).

**Figure 8 cells-13-00144-f008:**
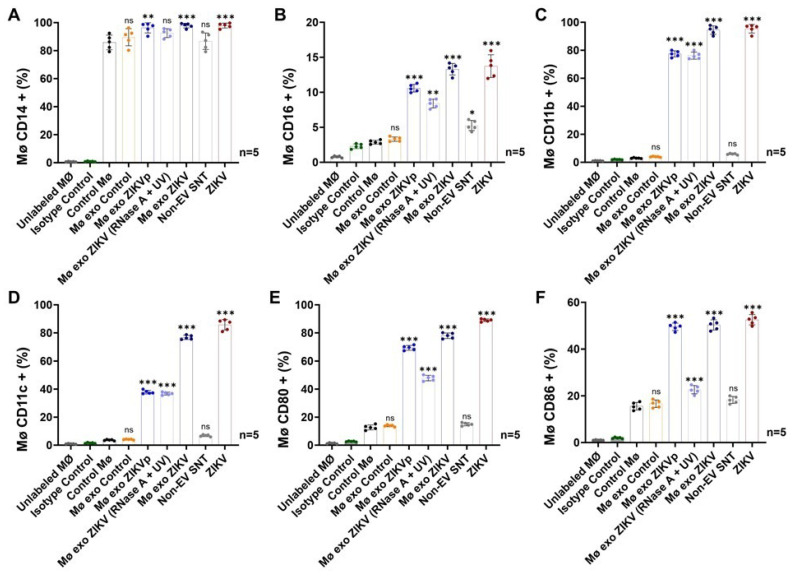
Cell-marker detection (by FACS) at 96 h post-stimulus with exosomes from ZIKV-infected Mø. (**A**) Percentages of Mø positive for CD14. (**B**) Percentages of Mø positive for CD16. (**C**) Percentages of Mø positive for CD11b. (**D**) Percentages of Mø positive for CD11c. (**E**) Percentages of Mø positive for CD80. (**F**) Percentages of Mø positive for CD86. Representative dot plots are shown in Appendix A. The percentages of Mø positive for CD14, CD16, CD11b, CD11c, CD80, and CD86 were compared with the Control Mø values by one-way ANOVA. Statistical significance is denoted as follows:* when *p* < 0.05, ** when *p* < 0.01, and *** when *p* < 0.0001. ns = no significance.

**Figure 9 cells-13-00144-f009:**
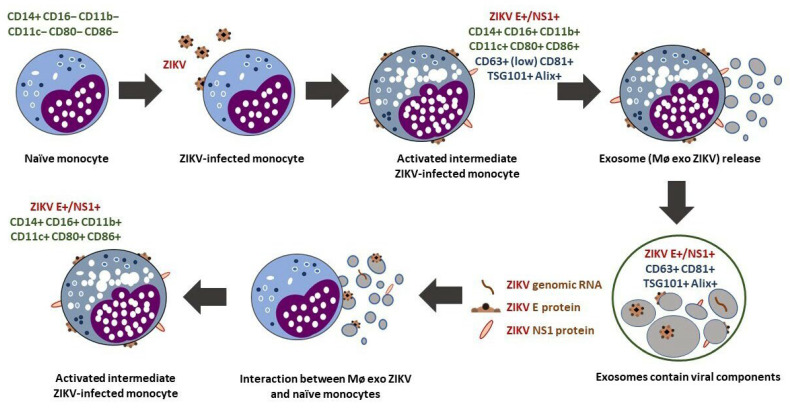
Zika virus-infected monocyte exosomes mediate cell-to-cell viral transmission (graphic description).

## Data Availability

The original contributions presented in the study are included in the article/Appendix A, further inquiries can be directed to the corresponding author.

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
