# Peer review of "Zika Virus-Infected Monocyte Exosomes Mediate Cell-to-Cell Viral Transmission"

_cells, 2024, doi:10.3390/cells13020144_

Round 1
Reviewer 1 Report
Comments and Suggestions for Authors
- Lanes 42-43 – „...infected-blood components transfusion, infected-organ transplantation, or laboratory exposure...). Please add references!
- Lane 109 – „1% 200 mM L-glutamine“. This is a confusing way for expression of concentrations (percentage vs molarity). Please indicate the final concentration of L-glutamine in mM
- Lane 109-110 – „1% antibiotic solution (penicillin 10,000 U/mL, streptomycin 10 mg/mL“. This is a confusing way for expression of concentrations (percentage vs IU/mg/mL). Please indicate the final concentrations for Penicillin in IU/ml, and for Streptomycin in mkg/ml.
- Lane 153 – What format was used (24-well, 12-well, 6-well)?
- Lanes 221-222 – The sentence „Cell pellets were discarded and the supernatants transferred to other tubes“ should be changed to „The supernatants were transferred to other tubes and cell pellets discarded“.
- Lanes 223-224 – The sentence „Cell debris pellets were discarded and the supernatants transferred to sterile ultracentrifuge tubes (25 × 89 mm, Beckman Coulter, Inc.) and centrifuged at 10,000x g at 4°C for 35 min“ should be changed to „The supernatants were transferred to sterile ultracentrifuge tubes (25 × 89 mm, Beckman Coulter, Inc.) and centrifuged at 10,000x g at 4°C for 35 min, and cell debris discarded“.
- Lanes 226-228 – „The exosome pellets were washed with PBS at 4 °C, incubated at rt for 30 min with constant agitation of 100 rpm, filtered using a 0.22 µm pore size, and centrifuged at 120,000x g for 70 min at 4 °C.“ – What PBS volume was used to resuspend the exosome pellet?
- Lane 253 – „fixated“ should be changed to „fixed“
- Lane 292 – „16,000 rpm“ – Please indicate in x g-s
- The diluent of BSA is not indicated (3%, 0.5% BSA). Water? PBS? DPBS?
- Lane 309 – Please indicate the concentration of Tween 20 in the washing solution
- Lane 594, Fig. 5H – RNA sizes are indicated in kbp-s. Correct please as ZIKV genomic RNA is a single-stranded molecule
- Lane 630 – „We found a visible band of approximately 11 kbp...“ – see previous comment
Reviewer 2 Report
Comments and Suggestions for Authors
This group previously showed that exosomes from ZIKV-infected mosquito cells carry viral genomic RNA and were infectious when exposed to naïve mosquito or mammalian cells. In this follow up study, the authors aim to determine the role of exosomes from ZIKV-infected monocytes in cell-cell transmission. Using various approaches, the authors show that ZIKV-infected monocytes also release exosomes that are infectious when exposed to naïve monocytes. The authors argue that the released exosomes can contribute to ZIKV persistence. The findings are interesting, but other ZIKV target cells [trophoblasts (nature Scient. Report. 2022. 12:7348); neuronal cells (Emerging Microbes and Infections. 2019. 8, 307-326)] also release infectious exosomes. Hence, it is unclear how significant the monocyte-derived exosomes are in spreading ZIKV or enabling viral persistence. In addition, there is too much redundancy in the data presented in all the figures. The quality of most of the figures is poor, hence hard to judge. Finally, the manuscript needs some editing.
Comments
The authors need to provide information on how the MFI was calculated in this study. In addition, they should address the significance of the MFI data in all the figures. The MFI data seem redundant in all the figures unless the authors can explain why they are critical.
Figures 3AB. If the monocyte CD63+ (%) cells are similar via FACS between control and ZIKV-infected cells (Fig. 3A), how can the monocyte CD63+ MFI be so different between control and infected cells? The authors should explain how the MFI data were generated. This comment applies to Fig. 3CD MFI.
In Figures 4 and 5, are viral particles in the media more infectious than the exosomes carrying viral elements? If so, what is the relative importance of the exosomes in viral transmission or spread?
In the finding from Figure 7, the authors state the following: "Therefore, these EV are efficient mechanisms for viral transmission by acting as Trojan vehicles that favor viral dissemination and infection in a receptor-independent manner". The authors may be correct, but the EVs are not exclusively made by ZIKV-infected monocytes nor do the authors argue that the EVs are the main route for release of infectious ZIKV from those cells.
In Figure 8, the authors suggest that exosomes from ZIKV-infected monocytes contain "stimuli" for naive monocyte differentiation and activation. While this is not explicitly stated, one would assume that the stimuli are ZIKV particles and/or viral genome. Given that ZIKV particles alone stimulate monocytes, a key advantage for the exosomes lies in stimulating cells via the viral genome. However, the ZIKV exosomes may not be exclusively derived from infected monocytes in the context of mammalian infection. They may come from infected endothelial cells, neuronal cells, placental cells, etc. If so, what makes the monocytes derived exosomes so important or significant in the context of mammalian infection?
Minor comments
Please give reference(s) for non-vectorial forms of ZIKV transmission.
Please provide references for exosomes released during ZIKV infection of other cells.
Comments on the Quality of English LanguageOverall, the manuscript is well written but it needs some editing.
